# Effects of Different Frying Oils Composed of Various Fatty Acids on the Formation of Multiple Hazards in Fried Pork Balls

**DOI:** 10.3390/foods12224182

**Published:** 2023-11-20

**Authors:** Mengyu Sun, Jin Wang, Jun Dong, Yingshuang Lu, Yan Zhang, Lu Dong, Shuo Wang

**Affiliations:** 1Tianjin Key Laboratory of Food Science and Health, School of Medicine, Nankai University, Tianjin 300071, China; 2120211402@mail.nankai.edu.cn (M.S.); wangjin@nankai.edu.cn (J.W.); yslu@nankai.edu.cn (Y.L.); yzhang@nankai.edu.cn (Y.Z.); 2Key Laboratory of Food Nutrition and Safety, Ministry of Education of China, Tianjin University of Science and Technology, Tianjin 300457, China; dongjun0827@163.com

**Keywords:** frying oil, unsaturated fatty acids, heterocyclic amines, polycyclic aromatic hydrocarbons, acrylamide, trans fatty acids

## Abstract

Oil oxidation products can react with food substrates to produce harmful substances, and oil saturation is closely related to oil oxidation in the process of frying. Therefore, the influence of the composition of fatty acids in oil on the formation of harmful substances in fried pork balls was explored. The five frying oils with the lowest unsaturated fatty acid (UFA) content, ranked in ascending order, were palm oil, peanut oil, soybean oil, corn oil, and colza oil (64.94%, 79.94%, 82.65%, 83.07%, and 92.26%, respectively). The overall levels of four harmful substances (acrylamide, polycyclic aromatic hydrocarbons, heterocyclic amines, and trans fatty acids) found in the oil used to fry pork balls followed a descending order: canola oil, corn oil, peanut oil, soybean oil, and palm oil (33.66 μg/kg, 27.17 μg/kg, 23.45 μg/kg, 18.67 μg/kg, and 13.19 μg/kg, respectively). This order was generally consistent with the trend in the content of UFAs. Therefore, the formation of harmful substances is closely related to the saturation of oil. Compared with other frying oils, soybean oil as a household oil produces relatively low amounts of harmful substances and has less negative impact on the quality (oil content, moisture content, and higher protein digestibility) of fried products.

## 1. Introduction

Meat is an important food source of essential nutrients and energy for the human body because of its rich content of protein, lipids, and minerals [1]. Frying is one of the traditional methods of processing meat. During the frying process, the meat products develop an attractive color and crispy taste, leading to the popularity of deep-fried foods [2]. However, meat products contain a large amount of oil after being fried. High oil intake is associated with multiple adverse health effects and may lead to a higher risk of disease, including hypertension, atherosclerosis, and cancer [3,4]. In addition, during the frying process, a series of chemical reactions occur in the matrix of frying meat products, such as the oil oxidation reaction, the Maillard reaction, and the oxidative degradation protein [5]. These chemical reactions lead to the production of hazardous substances in fried meat products, such as heterocyclic amines (HCAs), acrylamide (AA), polycyclic aromatic hydrocarbons (PAHs), and trans fatty acids (TFAs) [6,7,8,9].

AA is a hydrophilic compound that has been listed as a class 2A carcinogen by the International Agency for Research on Cancer [10]. The Maillard reaction is the main pathway for the formation of AA precursors from free asparagine and reducing sugars [11]. The effects of different frying oils (palm oil, coconut oil, canola oil, soybean oil, etc.) on fried foods with a high starch content and the conclusion that frying oil does affect the formation of AA have been thoroughly investigated [12,13]. Some studies have shown that AA could also be generated in fried meat products [7,14]. Evidence suggests that, although lacking reducing sugars, oxidative breakdown products of lipids, such as aldehydes and ketones, can also be involved in AA formation when carbonyls are present [15]. In addition, asparagine can form AA with the oil oxidation products octanal, 2-octanone, or 2, 3-butanedione [16]. Moreover, the oxidation capacity of frying oils varies due to their different contents of unsaturated fatty acids (UFAs). However, the effect of the UFA content in frying oils on the AA content in fried meat products has been poorly investigated.

PAHs belong to aromatic alkanes that contain more than two benzene rings in their molecules. PAHs are categorized as probable carcinogens (2A) by the IARC [17]. Proteins, lipids, and sugars in meat products can lead to the generation of PAHs, while oils are the key factor affecting PAH production during the frying process [18]. Nie et al., Ledesma et al., and Chen et al. studied the effects of saturated fatty acids (SFAs), monounsaturated fatty acids (MUFAs), and polyunsaturated fatty acids (PUFAs) on the formation of PAHs in flue gas and oil, respectively [18,19,20]. However, there are only a few studies on the effect of the fatty acid composition of oil on the amount of PAH residue in fried meat.

HCAs are a class of mutagenic and carcinogenic polycyclic aromatic compounds that are readily formed in high-protein meat [21,22]. HCAs are mainly produced through the Maillard reaction or pyrolysis of proteins and amino acids [23,24]. In addition, evidence suggests that oil oxidation can also affect the formation of HCAs [25], which is mainly because lipid oxidation products and amino compounds can form the precursor of HCAs and then promote the generation of HCAs [25]. However, there are no consistent results on the correlation between frying oil UFA content and HCA formation [21,23,25]. Tai et al. believe that highly saturated frying oil is more likely to form HCAs [21]. Ekiz et al. believe that when the oil contains more UFAs [25], it is easier to oxidize and promote the formation of HCAs.

TFAs are a kind of UFA that is widely used in fried food. It contributes not only to an increased risk of obesity but also to an increased risk of thrombosis, atherosclerosis, and type II diabetes [26,27]. Cui and Ayodeji studied the correlation between TFAs and frying oil in fried food and found that the formation of TFAs was significantly related to frying oil [28]. In addition, some studies have found that highly saturated frying oils are stable and not prone to isomerization, while highly unsaturated oils may produce TFAs due to their unsaturated double bonds [29].

In the frying process, frying oil is not only the heat transfer medium; it also participates in a series of chemical reactions in the food matrix that affect the formation of flavor [30]. Additionally, the fatty acid composition of the frying oil affects the quality of the fried food. Moreover, UFAs in frying oil, especially oleic acid and linoleic acid, undergo oxidative degradation reactions to generate aldehydes, ketones, and other substances during high-temperature frying, thus affecting the formation of harmful substances in food [31]. Studies on the effects of various lipids and fatty acid compositions on the formation of AA were limited to plant-based foods containing more carbohydrates, and the effects on the formation of harmful substances such as PAHs and HCAs were limited to simulated systems. This study, focused on fried pork balls prepared in an everyday family setting, filled the research gap and comprehensively analyzed the influence of frying oil types on the formation of multiple harmful substances (HCAs, AA, PAHs, and TFAs) in fried meat products. It provides some theoretical basis for choosing the proper frying oil for daily frying activities.

## 2. Materials and Methods

### 2.1. Experiment Materials

Pork tenderloin was bought at a supermarket in Tianjin, China. Palm oil (PO) was purchased from Julong Co. in Tianjin, China. Soybean oil (SBO), peanut oil (PNO), colza oil (CAO), and corn oil (CNO) were obtained from Luhua Co. in Yantai, China.

### 2.2. Experiment Reagent

The analytical-grade chemicals, such as ammonia, hydrochloric acid, sodium hydroxide, zinc acetate, and potassium ferrocyanide, were purchased from Aladdin Reagent Company in Shanghai, China. The HPLC-grade solvents, such as methanol, acetonitril formic acid, and n-hexane, were obtained from Merck Co. in Darmstadt, Germany. Pepsin, trypsin, and 22 standards (>99.99% purity), including Acrylamide, 13C3-acrylamide, Benzo[a]Anthracene (BaA), Chrysene (Chr), Benzo[b]Fluoranthene (BaF), Benzo[a]Pyrene (BaP), Harman (1-methyl-9H-pyridine[3,4-b]indoles), Norharman (H-pyridine[3,4-b]indole), PhIP (2-a-mino-1-methyl-6-phenyl-imidazolio[4,5-b]pyridine), A-aC (2-Amino-9H-pyrido[2,3-b]in-dol), MeAaC (2-Amino-3-methyl-9H-pyrido[2,3-b]ind-ol), IQ (2-Amino-3-methylimidazo-lio[4,5-f]-quinoline), IQx (2-amino-3-methylimidaz-o[4,5-f]quinoxaline), MeIQ (2-Amino-3,4-dimethylimidazo[4,5-ƒ]quinoline), MeIQx (2-amino-3, 8-dimethylimidazolium[4,5-f]quinoxaline), 4,8-DiMeIQx (2-amino-3,4,8-trime-thylimidazo[4,5-f]quinoxaline), 7,8-DiM-eIQx (2-amino-3,7,8-trimethylimidazo[4,5-f]qui-noxaline), Trp-P-2 (3-Amino-1-methyl-5-H-pyrido[4,3-b]indole), C16:1 9t, C18:1 9t, C-18:1 11t, C18:2 9t 12t, and C20:1 11t, were purchased from Sigma Co. (St. Louis, MO, USA).

### 2.3. Sample Pretreatment

The excess fat and tendons of the fresh pork were removed. The fresh pork was washed and made into minced meat. Each 20.0 g sample of ground pork was made into balls of the same shape and size, with no other substances added during the whole process.

The fryers were preheated to 180 °C. Then different frying oils (palm oil, soybean oil, peanut oil, colza oil, and corn oil) were used to fry the samples for 180 s at 180 °C. Samples were taken out, and the excess oil was absorbed from their surface. The samples were stored at −20 °C for testing.

### 2.4. Fatty Acid Composition in Original Frying Oil

The method detailed by Zribi et al. was referenced and slightly modified. A total of 60.0 mg of each oil was weighed, and then 4 mL of n-hexane and 0.2 mL of a potassium hydroxide-methanol solution were added [32]. The solution was mixed thoroughly and allowed to stand for 30 min. Then 1.0 g of anhydrous sodium sulfate was added. Finally, the mixture was separated at 1400× *g* for 5 min, and the supernatant was taken. GC-MS (7890B-5977A, Agilent, Santa Clara, California, USA) was used for analysis. The analysis conditions are shown in Table 1.

The area percentage obtained by the integral of each fatty acid peak was quantified. The results are expressed as a percentage of each fatty acid in the lipid component.

### 2.5. Determination of Basic Indicators

The oil content of the fried product was determined by the Soxhlet extraction method [33]. A 5.0 g sample of fried product was weighed and transferred to the filter paper cylinder. The filter paper cylinder was placed in the extraction cartridge of the Soxhlet extractor. After filtration, the petroleum ether was used for continuous reflux extraction for 6 h. At the end of the extraction, the petroleum ether was dried in a water bath. Then it was dried at 105 °C for 1 h and weighed again after cooling.

The moisture content of the fried product was determined by the direct drying method [34]. A 5.0 g sample was placed in a weighing flask and dried at 105 °C for 2 h, then taken out and cooled for 0.5 h before weighing.

### 2.6. Simulating Protein Digestibility In Vitro

The methods of Wen et al. and Escudero et al. were referenced and slightly modified. The 4.00 g samples of fried products were weighed in a centrifuge tube [35,36]. Then 16 mL of PBS was added and homogenized twice at high speed in an ice bath, and the pH value was adjusted to 2.0. Each sample was added to a 4 mL pepsin solution (0.0192 g: 15 mL 0.1 mol/L HCL), and the samples reacted in a constant temperature shaking table at 37 °C for 2 h. Then, the pepsin reaction was terminated by adjusting the pH value of the hydrolysate to about 7.5. The 4.0 mL of trypsin solution (0.288 g dissolved in 12 mL of 0.01 mol/L pH 7.0 PBS) was added to the hydrolysate, and the reaction was finished in the 37 °C constant temperature shaking table for 2 h. The reaction was terminated in a boiling water bath for 5 min. A total of 3 mL of anhydrous ethanol was added to the hydrolysate, and the hydrolysate was kept at 4 °C for 12 h. Then the hydrolysate was centrifuged (10,000× *g*, 20 min, 4 °C). After centrifugation, the supernatant was removed, and the precipitate was dried to a constant weight at 50 °C. The protein content in the samples before digestion and the residue from drying were determined by the Kjeldahl method [33]. The 1.00 g samples were accurately weighed and placed in a digestive tube. Two digestive tablets were added to each digestive tube, and 12 mL of concentrated sulfuric acid was added. Then the digestive tubes were placed in the digestive furnace and digested at 420 °C for 60 min. After finishing, the protein content of the samples was determined using the Hanon K1100F automatic Kjeldahl analyzer (Hanon Group, Jinan, China).

### 2.7. Determination of Acrylamide in the Fried Samples

The method detailed by Sanny et al. was referenced and slightly modified [37]. Firstly, the fried products were accurately weighed at 1.0 g and transferred into 50 mL centrifuge tubes. Secondly, 500 μL of ^13^C3-acrylamide (1 mg/L) and 9 mL of aqua pura were added. Then, the mixture was centrifuged for 30 min at 8500× *g*. Thirdly, 10 mL of n-hexane was used to remove oils from the supernatant, and the procedure was repeated three times. Next, after degreasing, 0.5 mL of zinc acetate and 0.5 mL of potassium ferrocyanide were used to remove the proteins. The purified samples were centrifuged at 1500× *g* for 10 min. After collecting the supernatant, it was passed through the C18 SPE column (Agilent, Santa Clara, CA, USA), and the effluent was collected. Then, 4 mL of methanol was used to elute the target. All the effluents were combined. The UPLC-MS/MS (Waters Ⅰ-Class LC-5500 QTRAP, Waters, Milford, MA, USA) was used to analyze the test solutions.

The C18 column (4.6 × 250 mm, 5 μm, Agilent, Santa Clara, CA, USA) was used with a temperature of 30 °C and a flow rate of 0.8 mL/min. Mobile phase A consisted of 0.1% formic acid, and mobile phase B consisted of methanol. The operating conditions of mass spectrometry were as follows: ion spray voltage, 5500 V; ion source gas, 165 psi; ion source gas, 265 psi; DP, 41 psi; CE, 16 psi. The transition values of acrylamide quantitation and multi-reaction monitoring (MRM) were *m*/*z* 72 → 55 and 72 → 44, respectively, and a 6-point linear calibration curve was established at 1 μg/L–100 μg/L (R^2^ > 0.9999). The detection limit and quantitative limit of the method were 0.5 μg/kg and 1.6 μg/kg, respectively. The recovery rate was stable in the range of 79.00–88.64%, and the relative standard deviations (RSDs) were 1.08–7.39%, which proved that the method was accurate and suitable for the determination of the acrylamide content in fried pork meatballs.

### 2.8. Determination of PAHs in the Fried Samples

The detection method detailed by Lee et al. was referenced and modified slightly [7]. The 5.0 g samples of fried products were weighed and transferred into 50 mL centrifuge tubes, and then 20.0 g of anhydrous sodium sulfate and 20 mL of cyclohexane-ethyl acetate (*v*/*v*: 1:1) were added. The mixture was extracted by ultrasound for 15 min and then centrifuged at 2400× *g* for 15 min. After extracting the supernatant and blowing it to near-dry using nitrogen, 5 mL of KOH-ethanol solution (1.5 mol/L), 4 mL of pure water, and 5 mL of n-hexane were added. The upper solution was extracted by centrifugation at 9400× *g* for 2 min. All extracts were purified using the UPAH dedicated solid-phase extraction column (Agilent, Santa Clara, CA, USA) (1 g of anhydrous sodium sulfate was added before use, and 5 mL of dichloromethane and 5 mL of n-hexane were successively added for activation). A total of 4 mL of n-hexane was eluted to remove impurities, and finally, 5 mL of a dichloromethane—ethyl acetate mixed solution was used for elution. The eluent was collected and concentrated to near-dryness by nitrogen blowing. Then, 0.5 mL of acetone-isooctane (*v*/*v*:1:1) was added. The purification solution was analyzed by the GC-MS (7890B-5977A, Agilent, Santa Clara, CA, USA). The GC-MS/MS conditions are shown in Table 1.

The selected ions of the PAHs are as follows: BaA (*m*/*z* 228,114,226), Chr (*m*/*z* 228,114,226), BaF (*m*/*z* 252,126,250), and BaP (*m*/*z* 252,126,250). The detection limits were 0.33 μg/kg–1.35 μg/kg, and the limits of quantification were 1.08 μg/kg–4.51 μg/kg. The recovery rate was stable between 79.30% and 103.47%, which proved that the method had good accuracy and precision and was suitable for detecting the contents of BaA, Chr, BaF, and BaP in fried pork meatballs.

### 2.9. Determination of HCAs in the Fried Samples

The method detailed by Wang [38] was referenced. The 6-point linear curve was established at 0.01 μg/L–20 μg/L (R^2^ > 0.999). The limits of detection were 0.0022 μg/kg–0.0172 μg/kg, and the limits of quantitation were 0.0072 μg/kg–0.0575 μg/kg. The recovery rate was stable between 50% and 88%, which proved that the method had good accuracy and precision and was suitable for the determination of HCAs in fried pork meatballs.

### 2.10. Determination of TFAs in the Fried Samples

The method detailed by Zribi et al. was referenced and slightly modified [32]. The 5.0 g samples of fried products were weighed and extracted with petroleum ether in continuous reflux for 6 h. After the reaction, the receiving flask was taken off, and the organic solvent was evaporated with a rotary evaporator (60 °C, 20 min). A total of 60 mg of the extracted fat was weighed, and 4 mL of isooctane was added to dissolve it before 0.2 mL of a 2 mol/L potassium hydroxide-methanol solution was added, and 1.0 g of sodium bisulfate was added to neutralize the excess potassium hydroxide. After swirling and mixing, the supernatant was centrifuged at 1400× *g* for 5 min. The result was analyzed by the GC (Agilent 7890B, Agilent, Santa Clara, CA, USA), with the GC analysis conditions shown in Table 1. The recovery rate was stable between 79.90% and 98.68%, which proved that the method had good accuracy and precision and was suitable for the determination of TFAs in fried pork meatballs.

### 2.11. Statistical Analysis

Microsoft Office Excel 2010 software was used to process the experimental data. SPSS (v.24.0, IBM Inc., Chicago, IL, USA) was used for a one-way analysis of variance (ANOVA). The results were expressed as the mean ± standard deviation (SD), and *p* < 0.05 was considered statistically significant. The analysis of correlations was performed using SPASS and Pearson’s correlation test, and *p* < 0.05 was considered statistically significant.

## 3. Results and Discussions

### 3.1. The Fatty Acid Composition in Original Frying Oil

Table 2 shows the fatty acid composition of the original frying oil. Palmitic acid and stearic acid were determined to be the main SFAs. Oleic acid was determined to be the main MUFA. Linoleic acid was determined to be the main PUFA. The proportion of the above four fatty acids to the total fatty acids in the five oils ranges from 94.86% to 98.99%. The UFAs in frying oil, especially oleic acid and linoleic acid, can produce oxidation products in the process of high-temperature frying, which affect the quality of fried products [31]. The relative content of oleic acid in PO was the highest; palmitic acid followed. Except for PO, linoleic acid and oleic acid were the two main fatty acids in the five oils, with linoleic acid being the most prevalent fatty acid in SBO and CNO and oleic acid being the most prevalent in PNO and CAO. The highest content of UFAs in the five frying oils, ranked from high to low, was CAO, CNO, SBO, PNO, and PO. The UFA content of CAO, SBO, and PO is consistent with the previous study that the UFA content of CAO is higher than that of SBO, and the UFA content of PO is the lowest [13]. The three oils with higher iodine values (IVs) were successively CNO, SBO, and CAO, followed by PNO and PO. Different from the UFA results, CAO mainly contained MUFAs, while CNO and SBO had more PUFAs, resulting in a lower IV for CAO than CNO and SBO.

### 3.2. The Basic Indicators

Moisture content, oil content, protein content, and protein digestibility are important indexes to evaluate the nutritional quality of food. Table 3 shows the oil content, moisture content, protein content, and protein digestibility of the fried products. When using SBO and PNO, the moisture content in fried products was higher than that in the other three groups (*p* < 0.05), and the CNO group had the lowest moisture content.

The type of frying oil affected the oil content of the fried products. When CNO and CAO were used, the oil content of the fried products was significantly higher than that of the other three groups (*p* < 0.05). The oil content in the PO group was the least. The oil content of the 5 groups, from low to high, was PO, PNO, SBO, CAO, and CNO, with contents of 2.56%, 3.83%, 4.02%, 4.95%, and 5.81%, respectively, which shows a similar trend to the IVs. This shows that the higher the unsaturation of the oil, the easier the oil is to soak into the food during frying, which is consistent with the results of previous research showing that the oil content of fried products was negatively correlated with oil saturation [32].

Protein was involved in the Maillard reaction, which then affected the quality of the food [39]. The type of frying oil affected the oil content of the fried products. When SBO was used as the frying oil, the protein content in the fried products was the highest (34.66%). The protein contents of the PO group and PNO group were 33.88% and 32.89%, respectively. When CNO and PNO were used as the frying oil, the protein content in the fried products was relatively low, at 31.87% and 31.68%, respectively.

The type of frying oil affected the protein digestibility of the fried products. When SBO and CNO were used, the protein digestibility of the fried product was better than that of PNO and CNO. However, when PO, which was low in UFAs, was used as the frying oil, the protein digestibility was lowest. The protein digestibility of the fried samples with the five oils is generally consistent with the IVs of the oils. This indicates that oils with higher unsaturation can produce more free radicals, which react with proteins and lead to decreased protein digestibility [40].

### 3.3. The Content of Acrylamide

Figure 1 shows the content of AA in the fried products. When PO was used to fry the pork meat balls, the AA content was the least, at only 4.78 μg/kg. When PNO was used, the AA content was significantly higher (*p* < 0.05), at 11.11 μg/kg. In between the two, CAO, SBO, and CNO had AA contents of 8.03 μg/kg, 6.23 μg/kg, and 5.78 μg/kg, respectively.

The type of frying oil affects the formation of AA in fried products. Although free asparagine and reducing sugar were the main precursors of AA [11], the current study shows that lipid oxidation also affects the formation of AA [41]. The reason for this is that in the process of high-temperature frying, oils undergo an oxidative pyrolysis reaction to generate oxides such as acrylic acid and acrolein, which are intermediates in the formation of AA [41]. Different frying oils have different fatty acid compositions, and the type of fatty acid affects the formation of aldehydes and other oxides [16], so the type of frying oil affects the formation of AA in fried products. Gertz et al. also found that different frying oils, such as PO, CAO, and sunflower oil, affect the formation of AA in fried French fries [41]. The content of AA in the PO group was lower than that in the CAO group, which matched the results of this study. Lim et al. also proved that there was a significant difference in the amount of AA produced in fried potato chips when using different frying oils (coconut oil, PO, CAO, and SBO) [12]. Raar et al. studied the effects of frying oil (PO, CAO, SBO, and sunflower oil) on AA and concluded that the frying oil can affect the formation of AA in fried products [13]. Their results showed that the formation of AA in CAO was higher than that in SBO, while the content of AA formed in the PO group was the lowest, which matched the research results of this study on the formation of AA in fried products.

### 3.4. The Content of PAHs

Table 4 shows the content of PAHs in fried products. The BaA content in the CNO group was the highest, followed by the CAO group, PNO group, and SBO group. The PO group, which has the highest SFA content, has the lowest BaA content. The content of Chr, BaP, and PAHs was the highest in the CAO group, which also had the highest UFA content, followed by the CNO group, PNO group, and SBO group, with the least in the PO group. BaF was not detected in the 5 groups of fried products. The results showed that the type of frying oil affected the formation of PAHs in fried products. This is because fatty acids were oxidized during frying at high temperatures to form hydroperoxide, which is an important precursor to the formation of PAHs [20,42]. What is noteworthy is that different frying oils have different fatty acid compositions, so the type of frying oil affects the formation of PAHs. At present, there are only a few studies on the effect of different frying oils on PAHs in fried products. An et al. studied the content of PAHs in various frying oils such as PNO, CAO, SBO, and PO [43]. The study showed that the content of PAHs in PNO was the highest with the increase in frying time, and the PAH content in CAO would increase significantly and be higher than in SBO and PO. Chen et al. studied the effects of hard fatty acids, oleic acid, linoleic acid, and linolenic acid, on PAHs in oil smoke and concluded that different fatty acids could affect the formation of PAHs, which also explained the results of this study, that is, the type of frying oil affects the formation of PAHs [20].

### 3.5. The Content of HCAs

Table 5 shows the content of HCAs in fried products. Twelve HCAs were determined, including Norharman, Harman, Trp-p-2, AaC, IQ, MeAaC, MeIQx, IQx, MeIQ, 4,8-DiMeIQx, PhIP, and 7,8-DiMeIQx. Among the fried products, Harman, Trp-p-2, MeAaC, and Norharman were mainly detected. There was a significant difference in the Norharman content and Harman content between each group (*p* < 0.05). The Norharman content and Harman content in the PNO group were the highest (0.89 μg/kg and 0.29 μg/kg, respectively), and the PO group had the lowest (0.31 μg/kg and 0.09 μg/kg, respectively). When CAO was used as the frying medium, the contents of MeAaC and Trp-P-2 in the fried products were higher than those in the other 4 groups (*p* < 0.05). The contents of MeAaC and Trp-p-2 in the CAO group were 0.41 μg/kg and 0.40 μg/kg, respectively. The total content of HCAs in the PNO group (1.89 μg/kg) and the CAO group (1.71 μg/kg) were significantly higher than that in the other 3 groups (*p* < 0.05). When PO, which is low in UFAs, was used as the frying oil, the total content of HCAs produced in the fried products was the lowest (1.17 μg/kg). The results indicate that the type of frying oil affects the formation of HCAs in fried products. At present, it has been reported that the reactive oxygen species produced by the oxidation decomposition of UFAs in frying oil at high temperature can induce the decomposition of the Maillard reactants into HCA intermediates, such as 1- and 3-deoxyketone, thus affecting the formation of HCAs [42,44].

### 3.6. The Content of TFAs

Figure 2 shows the content of TFAs in fried products. A total of 5 TFAs were determined, including C16:1 9t, C18:1 9t, C18:1 11t, C18:2 9t 12t, and C20:1 11t. When PO, SBO, and PNO were used, only 1 TFA was detected in the fried products. Two TFAs were detected in the CNO group, and three TFAs were detected in the CAO group. According to the total content of TFAs in the fried products, the order from high to low was: CAO, CNO, SBO, PNO, and PO. In addition, we also found that the content of TFAs in the fried products was affected by the UFA content of the frying oil. When frying oil with high unsaturation was used, the content of TFAs was higher. This is because, during frying, UFAs undergo isomerization reactions to form TFAs [45].

### 3.7. The Content of Total Hazards

Figure 3 shows the content of 4 kinds of harmful substances in fried products. We found that the type of frying oil significantly affects the formation of 4 kinds of harmful substances in fried products. When CAO was used as the frying oil, the total content of the 4 kinds of harmful substances was significantly higher than in the other 4 groups (*p* < 0.05). When PO, with the highest SFA content, was used as the frying medium, the total content of the 4 kinds of harmful substances was the least. The total content of the 4 kinds of harmful substances in fried products, from high to low, was CAO, CNO, PNO, SBO, and PO (33.66 μg/kg, 27.17 μg/kg, 23.45 μg/kg, 18.67 μg/kg, and 13.19 μg/kg, respectively). The results show that the type of oil affects the formation of 4 kinds of harmful substances in fried products. When frying oil with a high SFA content was used, less harmful substances were produced in the fried products.

### 3.8. Correlation Analysis

As shown in the above results, the use of different frying oils affects the quality of the fried products and the formation of AA, PAHs, HCAs, and TFAs. When frying oil with a high UFA content was used, the content of 4 kinds of harmful substances in fried products was higher than in frying oil with a high SFA content. Therefore, the correlation between the UFA content and the IV in frying oil and the formation of 4 kinds of harmful substances in fried products was analyzed (Figure 4).

The IV was significantly positively correlated with the MUFAs and PUFAs in the frying oil. The contents of MUFAs and PUFAs were higher when the IV was greater. In addition, there was a correlation between the IV of the frying oil and the quality of the food and the formation of harmful substances in the fried pork balls. There was a significant positive correlation between the IV of the frying oil and the total content of the 4 kinds of harmful substances, the oil content, and the protein content in the fried pork balls. That is to say, higher unsaturation in the frying oil leads to higher concentrations of harmful substances, oil content, and protein content in the frying pork balls. Although there was a significant positive correlation between the IV and UFA content in the frying oils, we also found that the effect of the frying oil IV on the formation of harmful substances was not consistent with that of the frying oil UFA content on the formation of harmful substances in the fried samples. This may be because the IV represents the number of unsaturated double bonds, and different UFAs contain different numbers of double bonds, which leads to different effects on the formation of harmful substances. As the precursor to the formation of hazards, it is very important to clarify the influence of fatty acid composition in oil on the formation of hazards. Therefore, we focused on the effect of the fatty acid composition of different oils on the formation of multiple hazards rather than focusing on oil saturation.

There is a certain correlation between the fatty acid composition of the frying oil and the formation of 4 kinds of harmful substances in fried products. The content of UFAs in the frying oil had a positive correlation with the formation of harmful substances in fried products. The total content of UFAs was significantly positively correlated with the content of 4 kinds of harmful substances (AA, PAHs, HCAs, and TFAs) (*p* < 0.05), with correlation coefficients of 0.78, 0.64, 0.77, and 0.86, respectively. In other words, when frying oil with a high UFA content was used, the contents of the 4 kinds of harmful substances were also high. In addition, the UFA content of frying oil had a significant positive correlation with the oil content and protein digestibility of the fried food.

There is no significant positive correlation between the contents of MUFAs, PUFAs, and UFAs in frying oil and the formation of AA in fried products. Kali et al. studied the effects of the fatty acid composition of different frying oils (SBO, sunflower oil, and virgin olive oil) on the formation of AA in baked potato chips, and it was concluded that the contents of MUFAs and UFAs in the frying oil have no significant effect, which was consistent with this study [46]. Frédéric et al. and Williams also proved that there was no significant correlation between the UFA content of frying oil and the formation of AA [47,48]. The main reason for this result may be that the food matrix is complex, and oil oxidation is not the only way to promote AA formation.

There is a significant positive correlation between the total content of UFAs in frying oil and the formation of PAHs in fried products (*p* < 0.05), and the correlation coefficient was 0.78. Fatty pyrolysis products such as methyl linolenic acid, methyl linoleate, methyl oleate, and methyl stearic acid were important precursors for the formation of PAHs [42], so the content of UFAs in frying oil affected the formation of PAHs in fried products, and there was a positive correlation. Nie et al. also studied and proved that there was a positive correlation between the content of UFAs and the formation of PAHs in the model system [18]. Under the same conditions, the group with the highest content of UFAs also had the highest content of PAHs.

As can be seen from Figure 4, the content of HCAs in fried products was significantly positively correlated with the total content of UFAs in frying oil (*p* < 0.05), with a correlation coefficient of 0.64. At present, the effect of frying oil on the formation of HCAs is not clear, and the results are not consistent. However, Barzegar et al. proved that there was a significant positive correlation between the formation of HCAs and oil content in the processing of meat products [49], that is, the content of HCAs in high-fat meat products was also very high. This could be due to the effect of oil on the heat conduction efficiency, which affects the HCAs, or it may be due to free radicals generated during fat oxidation that promote the pathway of HCA formation [50]. The UFA content of frying oil also has a significant positive correlation with the oil content of fried products (*p* < 0.05), which may explain the result that the UFA content of frying oil is positively correlated with the formation of HCAs. Sabally et al. and Bouchon et al. once pointed out that oil absorbed during frying was prone to build up on the surface of fried food, which was the main part of the formation of HCAs [51,52]. Tai et al. studied the effects of no frying oil and the use of lard, SBO, and coconut oil on the HCA content in fried fish patties and found that the total HCA content was at its lowest when no frying oil was used [21]. When lard, SBO, and coconut oil were used, the total HCA content increased significantly, which proved that the oil content affected the formation of HCAs. However, the research showed that the use of frying oils with a high SFA content produced higher HCA contents, which was inconsistent with the results of this study.

According to the results of the study (Figure 4), there is a significant positive correlation between the total content of UAFs in frying oil and the formation of TFAs in fried products (*p* < 0.05), and the correlation coefficient was 0.77. Cui et al. and Falade et al. had shown that the production of TFAs in fried products was closely related to frying oil [28,53]. Li et al. found that when frying with CAO, SBO, or PNO, the PNO group produced the least TFAs, which was consistent with the results of this study [54]. So far, there are few studies on the correlation between UFA content in frying oil and TFA content in fried products. But most studies have shown that the use of different frying oils affects the formation of TFAs in foods.

## 4. Conclusions

The type of frying oil affects the formation of harmful substances in fried products. AA and HCA contents were the highest in PNO-fried products (11.11 μg/kg and 1.89 μg/kg, respectively). The total contents of four kinds of harmful substances (AA, PAHs, HCAs, and TFAs) were the highest in CAO-fried products (8.67 μg/kg, 15.03 μg/kg, and 33.66 μg/kg, respectively). The contents of the four kinds of harmful substances in PO-fried products were the lowest, followed by SBO-fried products. There was a significant positive correlation between the total UFA content of the frying oil and the total amount of the four kinds of harmful substances (0.78, 0.64, 0.77, and 0.86, respectively). A significant positive correlation between UFA content and oil content and protein digestibility was found (0.73 and 0.55, respectively), and a significant negative correlation between the content of UFA and the protein content of the fried products (−0.53) was found.

Oil unsaturation had a significant impact on the production of hazardous substances. The soybean oil and colza oil used as frying oils in this study contained high levels of polyunsaturated fatty acids. Although polyunsaturated fatty acids have a high nutritional value, it should not be ignored that other unsaturated fatty acids can affect the formation of hazardous substances. The content of unsaturated fatty acids in colza oil is the highest compared to the other tested oils, and the amount of hazardous substances produced is also the highest. Considering both the nutritional value and the amount of hazardous substances produced, SBO as a frying oil had a relatively low content of harmful substances, and the effect on the quality of the fried products was also small. Therefore, we suggest that SBO be used as the primary frying oil in daily life.

## Figures and Tables

**Figure 1 foods-12-04182-f001:**
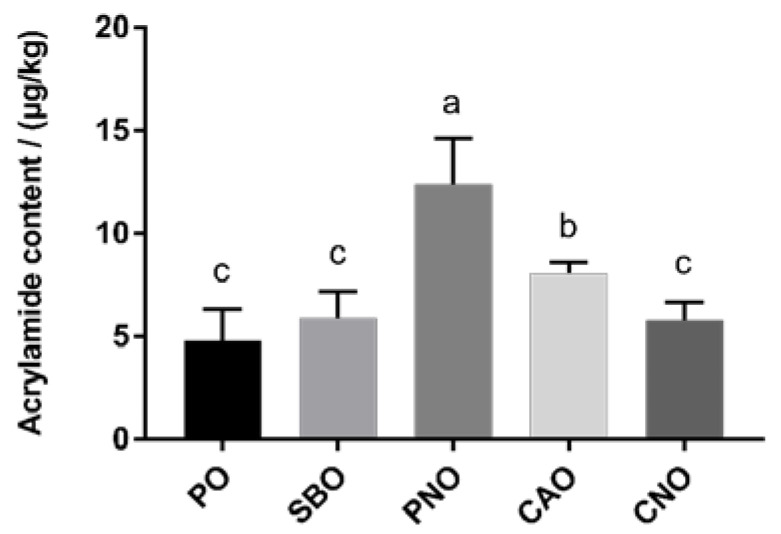
AA content in meatballs prepared with different frying oils. PO: palm oil; SBO: soybean oil; PNO: peanut oil; CAO: colza oil; CNO: corn oil. a–c: significant difference between groups.

**Figure 2 foods-12-04182-f002:**
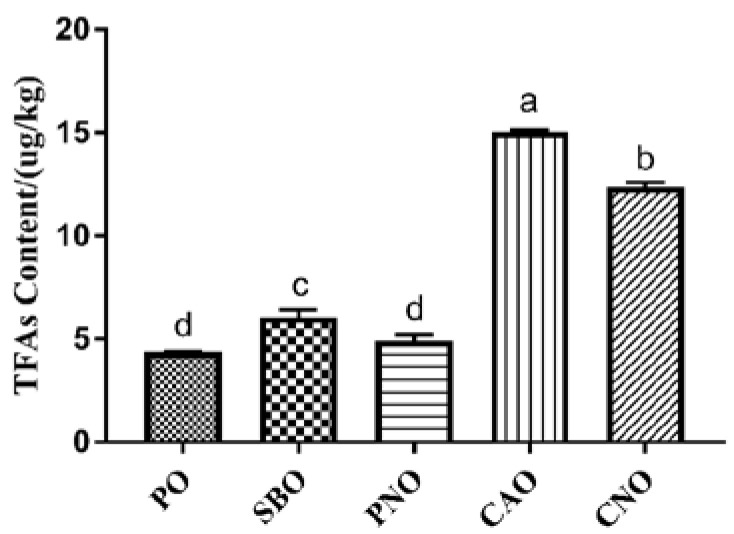
Content of TFAs in meatballs prepared with different frying oils. PO: palm oil; SBO: soybean oil; PNO: peanut oil; CAO: colza oil; CNO: corn oil. a–d: significant difference between groups.

**Figure 3 foods-12-04182-f003:**
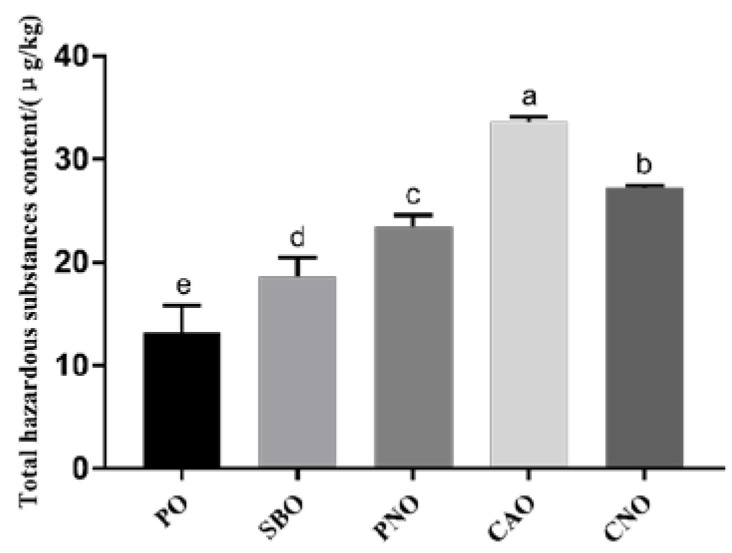
Total content of hazardous substances in meatballs prepared with different frying oils. PO: palm oil; SBO: soybean oil; PNO: peanut oil; CAO: colza oil; CNO: corn oil. a–e: significant difference between groups.

**Figure 4 foods-12-04182-f004:**
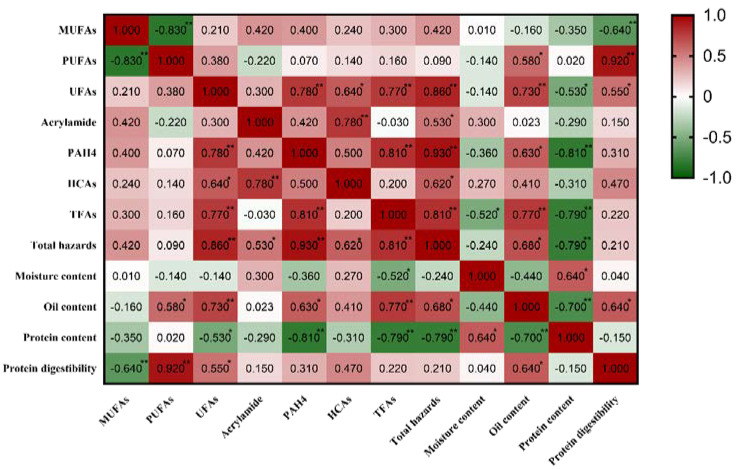
Correlation analysis of fatty acid composition with hazardous substances and basic indicators. * means *p* < 0.05, ** means *p* < 0.01.

**Table 1 foods-12-04182-t001:** Conditions for the gas chromatography analysis.

Item	Fatty Acid Composition	Polycyclic Aromatic Hydrocarbons	Trans Fatty Acids
Chromatographic column	HP-5 capillary column (0.25 μm × 0.25 μm ×30 m)		HP-88 capillary column (0.25 μm × 0.25 μm ×100 m)
Column temperature Box temperature	120 °C (4 min) → 10 °C/min, 175 °C (6 min) → 1 °C/min, 210 °C (5 min) → 10 °C/min, 230 °C (5 min)	80 °C (1 min) → 4 °C/min → 220 °C → (20 °C/min) → 280 min (10 min)	120 °C (4 min) → 10 °C/min, 175 °C (6 min) → 1 °C/min, 210 °C (5 min) → 10 °C/min, 230 °C (5 min)
Flow rate	1.5 mL/min	1.0 mL/min	1.0 mL/min
Inlet temperature	250 °C	320 °C	300 °C
Split ratio	5:1	-	1 μL

**Table 2 foods-12-04182-t002:** Relative contents of fatty acids in the original frying oil (%).

	PO	SBO	PNO	CAO	CNO
C16:0	30.06 ± 0.09 ^a^	11.85 ± 0.03 ^c^	11.28 ± 0.19 ^d^	4.76 ± 0.04 ^e^	14.15 ± 0.18 ^b^
C18:0	3.28 ± 0.01 ^c^	4.56 ± 0.00 ^a^	3.83 ± 0.04 ^b^	1.92 ± 0.10 ^d^	2.03 ± 0.01 ^d^
C18:1	51.90 ± 0.03 ^c^	30.95 ± 0.04 ^e^	52.70 ± 0.04 ^b^	68.16 ± 0.41 ^a^	31.54 ± 0.01 ^d^
C18:2	13.75 ± 0.01 ^d^	51.43 ± 0.01 ^a^	27.05 ± 0.02 ^b^	21.83 ± 0.54 ^c^	51.26 ± 0.04 ^a^
Other SFAs	1.73 ± 0.16 ^b^	0.94 ± 0.01 ^c^	4.96 ± 0.14 ^a^	1.07 ± 0.01 ^c^	0.76 ± 0.16 ^c^
Other UFAs	0.40 ± 0.01 ^b^	0.26 ± 0.01 ^c^	0.13 ± 0.00 ^d^	2.23 ± 0.02 ^a^	0.27 ± 0.05 ^c^
Other PUFAs	Nd	0.02 ± 0.00 ^c^	0.06 ± 0.00 ^a^	0.05 ± 0.00 ^b^	0.02 ± 0.01 ^c^
ΣSFAs	35.06 ± 0.06 ^a^	17.35 ± 0.04 ^c^	20.06 ± 0.01 ^b^	7.75 ± 0.15 ^d^	16.93 ± 0.00 ^d^
ΣMUFAs	51.20 ± 0.05 ^c^	31.20 ± 0.03 ^d^	52.83 ± 0.04 ^b^	70.39 ± 0.39 ^a^	31.80 ± 0.04 ^d^
ΣPUFAs	13.75 ± 0.01 ^d^	51.45 ± 0.01 ^a^	27.11 ± 0.02 ^b^	21.88 ± 0.54 ^c^	51.27 ± 0.04 ^a^
ΣUFAs	64.94 ± 0.06 ^d^	82.65 ± 0.04 ^b^	79.94 ± 0.02 ^c^	92.26 ± 0.16 ^a^	83.07 ± 0.00 ^b^
IV	75.73 ± 1.44 ^c^	130.13 ± 1.45 ^a^	111.13 ± 0.37 ^b^	114.70 ± 1.49 ^b^	134.22 ± 1.08 ^a^

a–e: significant difference between groups; Nd: not detected; PO: palm oil; SBO: soybean oil; PNO: peanut oil; CAO: colza oil; CNO: corn oil.

**Table 3 foods-12-04182-t003:** The basic indicators in meatballs prepared with different frying oils (%).

	PO	SBO	PNO	CAO	CNO
Oil content	2.56 ± 0.27 ^d^	4.02 ± 0.33 ^bc^	3.83 ± 0.859 ^c^	4.95 ± 0.62 ^ab^	5.81 ± 0.52 ^a^
Moisture content	64.17 ± 1.25 ^bc^	66.34 ± 0.38 ^a^	65.49 ± 0.14 ^ab^	64.22 ± 0.79 ^bc^	62.73 ± 0.89 ^c^
Protein content	33.88 ± 0.06 ^b^	34.66 ± 0.35 ^a^	32.89 ± 0.32 ^c^	31.87 ± 0.01 ^d^	31.68 ± 0.01 ^d^
Protein digestibility	19.10 ± 0.14 ^d^	30.65 ± 0.71 ^a^	27.80 ± 0.70 ^b^	24.44 ± 0.77 ^c^	31.53 ± 0.03 ^a^

a–d: significant difference between groups; PO: palm oil; SBO: soybean oil; PNO: peanut oil; CAO: colza oil; CNO: corn oil.

**Table 4 foods-12-04182-t004:** The content of PAHs in meatballs prepared with different frying oils (μg/kg).

	PO	SBO	PNO	CAO	CNO
BaA	1.26 ± 0.07 ^b^	1.30 ± 0.29 ^b^	1.58 ± 0.22 ^ab^	1.60 ± 0.18 ^ab^	1.85 ± 0.21 ^a^
Chr	1.03 ± 0.17 ^c^	1.24 ± 0.39 ^c^	2.73 ± 0.37 ^b^	3.91 ± 0.19 ^a^	2.75 ± 0.07 ^b^
BaF	Nd	Nd	Nd	Nd	Nd
BaP	2.32 ± 0.35 ^c^	2.38 ± 0.33 ^c^	2.80 ± 0.44 ^b^	3.91 ± 0.23 ^a^	2.77 ± 0.25 ^b^
PAHs	4.80 ± 0.07 ^d^	5.58 ± 0.25 ^c^	7.31 ± 0.34 ^b^	8.67 ± 0.17 ^a^	7.18 ± 0.33 ^b^

a–d: significant difference between groups; Nd: not detected; PO: palm oil; SBO: soybean oil; PNO: peanut oil; CAO: colza oil; CNO: corn oil.

**Table 5 foods-12-04182-t005:** Content of HCAs in meatballs prepared with different frying oils (μg/kg).

	PO	SBO	PNO	CAO	CNO
Norharman	0.31 ± 0.03 ^e^	0.74 ± 0.05 ^b^	0.89 ± 0.03 ^a^	0.51 ± 0.02 ^d^	0.60 ± 0.02 ^c^
Harman	0.09 ± 0.01 ^d^	0.23 ± 0.01 ^b^	0.29 ± 0.02 ^a^	0.17 ± 0.01 ^c^	0.22 ± 0.01 ^b^
Trp-P-2	0.25 ± 0.02 ^b^	0.13 ± 0.05 ^c^	0.21 ± 0.01 ^b^	0.40 ± 0.08 ^a^	0.24 ± 0.06 ^b^
MeAaC	0.25 ± 0.03 ^b^	0.18 ± 0.02 ^b^	0.15 ± 0.05 ^b^	0.41 ± 0.10 ^a^	0.19 ± 0.04 ^b^
Other HCAs	0.26 ± 0.01 ^a^	0.23 ± 0.04 ^a^	0.13 ± 0.04 ^b^	0.11 ± 0.04 ^b^	0.11 ± 0.05 ^b^

a–e: significant difference between groups; PO: palm oil; SBO: soybean oil; PNO: peanut oil; CAO: colza oil; CNO: corn oil.

## Data Availability

The data used to support the findings of this study can be made available by the corresponding author upon request.

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
