# Peer review of "Effects of Different Frying Oils Composed of Various Fatty Acids on the Formation of Multiple Hazards in Fried Pork Balls"

_foods, 2023, doi:10.3390/foods12224182_

Round 1

Reviewer 1 Report

Comments and Suggestions for Authors

This manuscript is on the effect of vegetable oil types used for frying on the content of some hazardous materials generated during the frying of meat.

It is an interesting and comperehensive research, but there are some points should be improved and clarified.

1- As it is clear from the fatty acid analysis (Table 2), the palm oil used in this manuscript, is palm olein rather than palm oil.

2- Introduction, line 63, ...flu gas and oil..., as it is clear from the references used (20 and 21), it was a misunderstanding. Please correct it.

3-line 204, PAH4, there is no such abbreviation, please correct it.

4- In figures, please write the full text of abbreviation or give a references for example "see the Table 1 for samples".

5- Title of the Figures should be complete and independent from the text.

6-Figure 2 should be as a Table to be clear and understandable. Also one chromatogram for this data could be better to have in the manuscript.

7- Among the oils, soybean and canola oils have the highest polyunsaturated fatty acids, but at the end soybean was the better oil, it should be discussed with a scientific literature review.

Comments on the Quality of English Language

In some places there was to "and", some times wrong word were used such as flu gas and so on, which need more careful check.

Author Response

Dear reviewer,

Thank you for your patience and kind work on our manuscript entitled “Effects of different frying oil composed with different fatty ac-ids on the formation of multiple hazards in fried pork balls” (foods-2717558). The comments are helpful in improving the manuscript. The amendments in the revised manuscript have been yellow-highlighted. Our responses to the comments are as follows.

1- As it is clear from the fatty acid analysis (Table 2), the palm oil used in this manuscript, is palm olein rather than palm oil.

Response: Thank you for your professional comment. “Palm oil” is a generic name for oil extracted from the oil palm genus, which can be classified according to its different extraction parts in the oil palm. As the fatty acid composition of untreated vegetable oil and the types of triglycerides are mixed, it is difficult to meet the nutritional needs of consumers and the application is limited, usually commercial palm oil will be treated by separation and extraction technology according to its use, such as palm olein you mentioned, because of its better oxidation stability, it is often used as frying oil. In this study, we selected commercially marketed “palm oil”, which was more similar to palm olein through fatty acid analysis, which was also in line with the frying processing method in the study.

2- Introduction, line 63, ...flu gas and oil..., as it is clear from the references used (20 and 21), it was a misunderstanding. Please correct it.

Response: We were sorry for our careless mistakes. We have corrected the accuracy of the references cited here, then the serial numbers of the references in the manuscript were also corrected. (line 63 in revised manuscript)

3-line 204, PAH4, there is no such abbreviation, please correct it.

Response: Thanks for your careful checks. We have corrected all PAH4 to PAHs throughout the manuscript. (line 205 in revised manuscript)

4- In figures, please write the full text of abbreviation or give a reference for example "see the Table 1 for samples".

Response: Thank you for your reminding. A note for the abbreviation is listed after each Figure’s legend.

5- Title of the Figures should be complete and independent from the text.

Response: Thank you for your comment and we have corrected the problem.

6-Figure 2 should be as a Table to be clear and understandable. Also one chromatogram for this data could be better to have in the manuscript.

Response: Thank you for your suggestion. We strongly agree that it is more intuitive to use a table to display the HCAs content. (Table 5 in revised manuscript)

In addition, the content of HCAs was determined by LC-MS. The chromatogram can explain the separation of HCAs, we will provide the chromatogram of the sample in the supplementary material.

7- Among the oils, soybean and canola oils have the highest polyunsaturated fatty acids, but at the end soybean was the better oil, it should be discussed with a scientific literature review.

Response: Thank you for your professional comment. The purpose of this study was to explore the correlation mechanism between oil saturation and the formation of harmful substances in fried meat products. Our study found that the content of unsaturated fatty acids was significantly positively correlated with the content of HCAs, PAHs and TFAs. Therefore, oil unsaturation had a significant impact on the production of hazardous substances. The frying oil of soybean oil and colza oil used in this study contains high levels of polyunsaturated fatty acids. Although polyunsaturated fatty acids have high nutritional value, we cannot ignore that other unsaturated fatty acids, which can also affect the formation of harmful substances. The content of unsaturated fatty acids in colza oil is the highest, and the content of hazardous substances produced is also the highest. Considering both the nutritional value and the content of hazardous substances produced, we conclude that soybean oil is more suitable for use as frying oil. In the Conclusion part of the revised manuscript, the conclusion that soybean oil is more suitable as frying oil was explained in detail.

Reviewer 2 Report

Comments and Suggestions for Authors

Main Comments:

1.The conclusion that SBO is useful as an oil for daily use lacks adequate explanation, especially considering that data from Fig. 1 to Fig. 4 appear to indicate that PO is a better oil. This discrepancy should be addressed and explained more thoroughly.

2.The paper should provide more information on the characteristics of the oils used in the experiment and discuss previous research related to them in the introduction.

3.The introduction should include references to past research examples, especially regarding oils used in similar studies, such as SBO and PO.

Minor Comments:

1.Ensure that figure legends are properly written.

2.Rewrite sentences starting with "And" at line 340, 406, and 420 to enhance reader comprehension.

3.Consider writing out the full names for abbreviations such as HAAs and PAH4 on line 95, or clarify their correctness if they are incorrect.

Comments on the Quality of English Language

 Minor editing of English language required.

Author Response

Dear reviewer,

Thank you for your patience and kind work on our manuscript entitled“Effects of different frying oil composed with different fatty ac-ids on the formation of multiple hazards in fried pork balls” (foods-2717558). The comments are helpful in improving the manuscript. The amendments in the revised manuscript have been yellow-highlighted. Our responses to the comments are as follows.

1.The conclusion that SBO is useful as an oil for daily use lacks adequate explanation, especially considering that data from Fig. 1 to Fig. 4 appear to indicate that PO is a better oil. This discrepancy should be addressed and explained more thoroughly.

Response: Thank you for your professional comment. From the perspective of food processing, PO have better oxidation stability, it was usually be used as oil for industrial processing of food. From the perspective of nutrition, other kinds of oils have higher content of unsaturated fatty acids and higher nutritional value. In addition to PO, SBO has a higher unsaturated fatty acid content, and at the same time, fewer harmful substances are produced during frying. Therefore, from the point of view of home cooking, SBO is a better frying oil. In the conclusion part of the revised manuscript, the conclusion that SBO is more suitable as frying oil in daily life was explained in detail.

2.The paper should provide more information on the characteristics of the oils used in the experiment and discuss previous research related to them in the introduction.

Response: Thank you for your comment. The purpose of this study was to explore the correlation mechanism between oil saturation and the formation of harmful substances in fried meat products. The fatty acid composition of the raw oil in the study was analyzed in detail, which can greatly assist our research objectives. Existing literatures reports on oil saturation and hazard formation are also discussed in detail in the Introduction. (lines 46-48, 60-63, 68-75 in revised manuscript)

3.The introduction should include references to past research examples, especially regarding oils used in similar studies, such as SBO and PO.

Response: Thank you for your suggestion. At present, there are few studies on the correlation mechanism between the oil saturation of different frying oils and the formation of multiple hazards. Some studies only discuss the influence on the production content of certain hazards, which was discussed in the Introduction (references [12, 13, 18-20, 21, 23, 25]). In our study, we hope to provides a more comprehensive correlation of the influence of oil saturation on the formation of multiple hazards.

4.Ensure that figure legends are properly written.

Response: Thank you for your reminding. The figure legends were rewritten in the revised manuscripts. (lines 311, 367, 382, and 393)

5.Rewrite sentences starting with "And" at line 340, 406, and 420 to enhance reader comprehension.

Response: Thank you for your reminding. The sentences were rewritten as follow:

The content of MeAaC and Trp-p-2 in CAO group were 0.41 μg/kg and 0.40 μg/kg, respectively. (line 343 in revised manuscript)

The content of UFAs in frying oil was a positive correlation with the formation of harmful substances in fried products. (line 413 in revised manuscript)

Frédéric et al. and Williams also proved that there was no significant correlation between the UFA content of frying oil and the formation of AA [51,52]. (line 427 in revised manuscript)

6.Consider writing out the full names for abbreviations such as HAAs and PAH4 on line 95, or clarify their correctness if they are incorrect.

Response: Thank you for your checks. The abbreviations of HAAs and PAH4 were corrected to HCAs and PAHs, and the four abbreviations were referred in the previous, but it should follow “multiple harmful substances” in text, please see line 95.
